# Oncolytic Viral Therapy in Osteosarcoma

**DOI:** 10.3390/v16071139

**Published:** 2024-07-16

**Authors:** Thomas Karadimas, Thien Huong Huynh, Chloe Chose, Guston Zervoudakis, Bryan Clampitt, Sean Lapp, David Joyce, George Douglas Letson, Jonathan Metts, Odion Binitie, John E. Mullinax, Alexander Lazarides

**Affiliations:** 1Morsani College of Medicine, University of South Florida Health, Tampa, FL 33602, USA; huynh27@usf.edu (T.H.H.); cchose@usf.edu (C.C.); baclampitt@usf.edu (B.C.); seanlapp@usf.edu (S.L.); 2Sarcoma Department, Moffitt Cancer Center, Tampa, FL 33612, USA; guston.zervoudakis@moffitt.org (G.Z.); david.joyce@moffitt.org (D.J.); douglas.letson@moffitt.org (G.D.L.); jonathan.metts@moffitt.org (J.M.); odion.binitie@moffitt.org (O.B.); john.mullinax@moffitt.org (J.E.M.); alexander.lazarides@moffitt.org (A.L.)

**Keywords:** oncolytic viral therapy, oncolytic viruses, osteosarcoma

## Abstract

Primary bone malignancies, including osteosarcoma (OS), are rare but aggressive. Current OS treatment, involving surgical resection and chemotherapy, has improved survival for non-metastatic cases but remains ineffective for recurrent or metastatic OS. Oncolytic viral therapy (OVT) is a promising alternative, using naturally occurring or genetically modified viruses to selectively target and lyse cancer cells and induce a robust immune response against remaining OS cells. Various oncolytic viruses (OVs), such as adenovirus, herpes simplex virus, and measles virus, have demonstrated efficacy in preclinical OS models. Combining OVT with other therapeutics, such as chemotherapy or immunotherapy, may further improve outcomes. Despite these advances, challenges in reliability of preclinical models, safety, delivery, and immune response must be addressed to optimize OVT for clinical use. Future research should focus on refining delivery methods, exploring combination treatments, and clinical trials to ensure OVT’s efficacy and safety for OS. Overall, OVT represents a novel approach with the potential to drastically improve survival outcomes for patients with OS.

## 1. Introduction

Primary malignancies of bone are rare, representing only 0.2% of all cancer diagnoses and accounting for 5% of childhood cancer diagnoses [1]. The most common of these is osteosarcoma (OS), a bone tumor in which the tumor cells produce immature bone, known as osteoid [2]. Despite its rarity, osteosarcoma is particularly aggressive with 10 to 20 percent of patients exhibiting evidence of macroscopic metastatic disease at presentation. Metastases most commonly occur in the lungs, followed by other bones [3].

While a range of treatment strategies exist for OS, conventional treatment for localized disease involves the combination of wide surgical resection with neoadjuvant and/or adjuvant chemotherapy. The chemotherapy regimen typically includes doxorubicin and cisplatin, with or without subsequent high-dose methotrexate [4]. For nonmetastatic OS, the introduction of multiagent chemotherapy and improvement of surgical techniques initially increased 5-year survival three-fold to 65–70%, making radical resection with limb salvage possible in greater than 90% of cases [5,6]. In stark contrast, survival outcomes for patients with recurrent or metastatic osteosarcoma remain poor. Despite the initial success of these treatment advances, overall survival rates for OS have not improved appreciably in nearly four decades, necessitating the exploration of novel therapies to advance survival outcomes in this disease [7].

Both targeted gene therapy and immunotherapy have shown recent promise for the treatment of cancer. However, the genetic heterogeneity and immunosuppressive microenvironment of OS have made the utilization of these treatment modalities extremely difficult [7]. Oncolytic viral therapy (OVT) is a new cancer immunotherapy that utilizes naturally occurring or genetically modified viruses that specifically target malignant cells. The role of oncolytic viruses (OVs) in the treatment of cancer is two-fold: these viruses induce oncolysis and activate immune system cells, triggering both innate and adaptive immune responses [8]. OVs have been shown to modify the tumor microenvironment and target tumor vasculature [9,10]. A wide range of OVTs are currently being evaluated for the treatment of various cancers including adenovirus, vaccinia virus, measles virus, parvovirus, and several others [11]. To date, only Talimogene laherparepvec (T-VEC), a herpes simplex virus that is delivered through an intralesional injection, has been granted FDA approval for the treatment of metastatic melanoma [12]. The diverse and robust immune response elicited by OVs highlights their role as a powerful therapeutic tool. Our knowledge on the use of OVT in osteosarcoma is currently sparse but has garnered increasing attention as a treatment option in recent years. This review seeks to compile and summarize the current findings regarding OVT within the context of OS.

### Proposed Mechanism of Action of OVT in Osteosarcoma

The general mechanism by which OVT works is via selective tumor cell infection and lysis with subsequent formation of antitumor immunity (see Figure 1 below). OSs are characterized by aberrant cellular proteins and pathways, such as retinoblastoma tumor suppressor (Rb) telomerase, thymidine kinase, Ras, and many others. Additionally, OSs have been shown to modify the local environment to evade the immune system [13]. These features contribute to the aggressiveness and pathologic fitness of OS by facilitating unchecked cell growth. Alternatively, these same features allow unchecked viral replication, which is the basis for the efficacy and selectivity of OVT. Once infected, OVs replicate and lyse tumor cells, inducing immunogenic cell death (ICD). ICD corresponds to the release of damage-associated molecular patterns (DAMPs), pathogen-associated molecular patterns (PAMPs), and tumor antigens. These molecules induce cytokines which trigger a strong immune response towards tumor cells locally, leading to their destruction. OS-specific antigens can also induce a systemic adaptive immune response, priming the immune system against all OS cells in the body [14,15]. Additionally, OVs can be further modified to alter the OS tumor microenvironment in a way that promotes local immune infiltration and limit tumor growth [13].

## 2. Therapeutic Potential of OVT in OS

Multiple studies have explored the application of OVT for the treatment of OS, both as a monotherapy and in combination with other cancer treatments. These studies include promising preclinical evaluations of OVT’s effects in various treatment settings (see Table 1 below).

### 2.1. Monotherapies

Regarding OVT monotherapy for OS, there have been multiple viruses studied to date. Delta-24-ACT is an oncolytic adenovirus (OAd) that is modified to express immune costimulatory ligand 4-1BB (4-1BBL). This ligand, when expressed on virally infected cells, interacts with T cells to promote the survival and expansion of activated T cells, improving targeting of local OS and potentiating a systemic immune response [16]. In a human OS cell line, Delta-24-ACT killed OS cells and triggered the production of damage-associated molecular patterns (DAMPs), a proxy for immunogenic cell death (ICD). In vivo in a murine model, Delta-24-ACT had antitumor effects against both the primary tumor and metastases, and significantly increased survival time. Most notably, all long-term responders developed an anti-OS memory immune response that was effective against rechallenge with the OS cell line. Finally, this OVT exhibited no toxicity [16].

Another OAd is VCN-01, with multiple modifications to improve cancer cell infectivity and tumor microenvironment penetrance. The first is a RGDK motif in the heparin sulphate-glycosaminoglycans (HSG)–binding domain KKTK of the fiber shaft, which readily binds HSGs and integrins that are highly expressed on malignant cells, resulting in greater affinity and specificity for cancer cells. The second is hyaluronidase, which degrades extracellular matrix hyaluronic acid, disturbing the tumor microenvironment and allowing greater immune infiltration. Past research has supported VCN-1 as being a potent antitumor virus both in human OS cell lines and in mice with tibial OS or metastatic OS with no toxicity [17].

Shifting to another OVT for OS, vaccine strain measles virus (MV) has been shown to be a promising species for OS viral therapy. Vaccine strain MV enters cells primarily through CD46, which is highly expressed in tumors, making it specific for cancer cells. MV also has potent anti-OS effects in human OS cell lines as well as decreasing tumor size and extending median survival time in mice with tibial and metastatic OS [21].

An additional virus that has been studied for OVT in OS is protoparvovirus H-1 (H-1PV). This virus selectively infects highly proliferative cells, such as OS cancer cells, because its life cycle depends on host cell factors involved in proliferation and replication [27]. In the context of OS in human cell lines, H-1PV had strong anti-OS effects, with no cytotoxicity to normal cells [22]. Although H-1PV has not been explored for OS in vivo, in mice with Ewing’s sarcoma, H-1PV also showed an antiproliferative effect, but did not improve survival [28].

Reovirus (RV) is another virus that has been trialed in OVT for OS. RV is a naturally occurring virus with preferential replication in cancer cells with activated Ras signaling, such as in OS. RV inhibits growth and viability of human OS cells in vitro and prevents progression of OS tumors in mice [23]. The only other setting in which RV has been studied for OS is in canine cell lines, which were minimally susceptible to RV infection [24]. In the context of cancer patients, RV has been shown to be safe and well tolerated [25].

A further option for OVT in the context of OS is vesicular stomatitis virus (VSV). Prior research has shown a significant oncolytic effect of VSV both in vitro and in vivo in mice with improvements in short-term but not long-term survival [29,30]. This has prompted modification of VSV to encode antitumor micro-RNA143, which is downregulated in OS and may contribute to its pathogenicity. Injection of miRNA143 decreases metastatic potential of OS and VSV expressing miRNA143 improves specificity in other cancers [31,32]. In OS models in vitro, VSV-miRNA143 had significant cytotoxic and antimigration effects, and these effects were much greater than unmodified VSV [26]. Additionally, VSV is a relatively benign virus in humans and is unlikely to have severe cytotoxic effects on normal cells [33].

### 2.2. Combination Therapies

OVT may not be able to fully eliminate OS when applied alone. Therefore, combining OVs with other therapies, such as chemotherapy and immunotherapy, offers a compelling strategy to further improve oncologic outcomes. One such example of combinatory therapy combines an OAd (dlE102) with granulocyte-colony stimulating factor (G-CSF). OAds target cells with a defective Rb pathway, which is commonly impaired in OS tumors and can be administered via virus-infected human mesenchymal stem cells (MSCs) with suggestion of improved tumor tropism [34]. G-CSF is traditionally used to manage neutropenia in patients receiving chemotherapy [35,36] and has been reported to enhance immune cell infiltration into tumors and to provide more effective responses to immunotherapies [37,38,39]. In vitro and in vivo studies utilizing OAd dlE102-MSCs to treat OS have shown favorable results. Murine models of OS treated with OAds experience reduced tumor size, as well as increased survival when combined with G-CSF. The immunologic profiles of mice treated with OAd-MSCs were comparable to OS patients with good prognoses [18]. Additionally, OAd-MSCs that have been previously trialed for neuroblastoma and other relapsed/refractory solid tumors demonstrated an excellent safety profile [40,41,42].

Another combination therapy that has been studied is doxorubicin and OAd that is modified to express p53 (OBP-702), theorized to produce a more potent oncolytic effect through the p53-apoptotic pathway [43,44]. A previous phase I clinical trial of a similar OAd without p53 expression showed promising safety and antitumor effects in the context of other malignancies [45,46,47]. In human OS cell lines, with a set dosage of DOX, addition of OAd OBP-702 decreased the number and viability of both DOX-sensitive and resistant OS cells in a dose-dependent manner. Furthermore, using a DOX-resistant OS tumor model in mice, combination DOX and OBP-702 significantly mitigated tumor growth, while treatment with either treatment alone did not have this effect [19].

Reovirus (RV), described earlier under monotherapies, has also been looked at in combination with cisplatin. In mice with OS, RV prevents growth of tumors when applied alone, but when combined with cisplatin, these mice showed a significant reduction in tumor size [23].

An additional combination therapy involves use of a modified herpes simplex virus, OV HSV1716, plus anti-PD-1. HSV1716 (Seprehvir) is an HSV-1 OV with deletions in both copies of the neurovirulence gene RL1 (encoding virulence factor γ_1_34.5), making it selectively target cancer cells [20]. PD-1 is an immune checkpoint receptor highly expressed on T cells. The ligand to this receptor, PD-1-ligand (PD-L1), is expressed on a variety of cells throughout the body. Binding of these molecules leads to T cell inactivation, thereby preventing tissue damage and maintaining self-tolerance. Although necessary to keep immune responses in check, malignant cancers, including OS, hijack this pathway by expressing high levels of PD-L1 to evade the immune system [48]. Many immunotherapies such as anti-PD-1 and anti-PD-L1 target this pathway but are ineffective in treating OS alone [49]. In vivo, HSV1716 plus anti-PD-1 has differential survival effects dependent on the line of OS. In mice with less-immunogenic OS tumors, HSV1716 and anti-PD-1 alone or in combination did not affect tumor growth or survival. However, in mice with highly immunoreactive OS tumors, combination therapy delayed tumor growth and prolonged survival. Finally, HSV1716 (Seprehvir) and anti-PD-1 (Pembrolizumab) have independently been shown to be safe in prior studies, although further studies are necessary to validate their safety in a combined approach [50,51,52].

## 3. Current Challenges and Barriers to Implementation

Despite positive preclinical and clinical data supporting OVT, various challenges regarding the utility of preclinical trials, clinical safety, efficacy, delivery, and regulation must be addressed prior to the adoption of OVT as a mainstay treatment of OS.

### 3.1. Preclinical Challenges

Regarding the preclinical evaluation of OVs, many viruses, such as varicella-zoster virus, have restrictive species tropism, limiting the utility of in vivo animal models [53]. Additionally, viruses such as herpes simplex virus type 1 infect and propagate through different species in a distinct manner, undermining the results from animal models [54,55,56,57]. Immunocompromised mice can host human OS, overcoming some limitations of syngeneic models. However, this also compromises the applicability of these studies, as OVT relies on an immunocompetent host [58]. In addition, most preclinical OV studies plant tumors superficially for ease of administration, but this can result in deviations from the typical tumor microenvironment seen in patients with OS [59]. Orthotopic mouse models are more ideal for recreating the microenvironment of clinical cases but are more labor intensive and time consuming for tumor implantation and virus administration. Animal models are incredibly useful for determining efficacy and safety of OVT, but these barriers highlight the limitations of such models for extrapolation of data to humans [12].

### 3.2. Safety: Off-Target Effects, Toxicities, Mutation, Viral Transmission

As live viruses, OVs present concerns for off-target effects, unexpected toxicities, mutation, recombination with wild-type viruses in the environment, and transmission to others [12,14]. Additionally, administration of some OVs is associated with fever, chills, nausea, vomiting, arthralgia, among many other symptoms resulting from a systemic immune response [60]. These adverse outcomes are difficult to predict, making the ability to eliminate OVs a crucial safety component of therapy [16,40,41,42,50]. This feature makes oncolytic HSVs, such as HSV1716, an attractive option for OS. These OVs retain the enzyme thymidine kinase, making them easily neutralizable and limiting the potential for serious adverse events [42,50]. The risk of transmission to healthy individuals has been mitigated through the development of multiple protocols to aid clinicians and patients in proper storage, handling, administration, and sterilization techniques [61,62,63]. Additionally, there has been no evidence of transmission of the FDA-approved OV T-VEC to healthy individuals [12]. Nonetheless, short- and long-term safety with OVT remains a concern, reflected in clinical trials where immunocompromised patients and those with active viral infections are excluded [15].

### 3.3. Host vs. Virus

Another challenge of OVT is the immune system’s ability to recognize and neutralize the OV before it can produce a therapeutic effect. First, the innate immune system is exceptional at targeting viruses and may quickly eliminate the virus before it can reach OS cells in significant quantities [64,65,66]. Additionally, the adaptive immune system’s neutralizing antibodies also diminishes the efficacy of OVs. More than half of humans already have antibodies against HSV-1, measles, and reovirus, and in response to OVT, antibodies are rapidly generated, limiting the window and dose range in which therapy is effective [67,68,69,70,71,72]. Due to these factors, high and frequent administered viral doses required to overcome these antibodies lead to administration and safety concerns [13,68]. Immunosuppressive agents that dampen the immune response, such as cyclophosphamide [73], or engineering viruses to evade immune detection may combat this. However, these approaches come with their own pitfalls such as increased susceptibility to infection, treatment toxicity, and potential negation of the immunogenic and antitumor effect of OVTs [12]. This necessitates a delicate immunologic balance to ensure the efficacy of OVT for OS while retaining patient safety.

### 3.4. Delivery Method

The delivery mechanism of OVs presents another challenge to its application in OS. Due to a systemic immune response limiting viral spread, intratumoral injection is the most advantageous route for administering OVT, and this is the route currently being used for the only FDA-approved OVT, T-VEC [74]. Intratumoral injections are ideal for superficial and small tumors, but this method comes with the risk of damaging implicated structures with large tumors or deep metastases, such as to the lungs [13,75]. Other routes, such as intravenous administration can reach virtually any metastasis; however, the viral load at these locations is often decreased due to significant viral destruction by the host. This has prompted exploration of specific viruses, such as vaccinia, that when enveloped is not neutralized by antibodies, and can reach distant organs with high viral loads [76,77]. Additionally, a delivery system utilizing nanoparticles is a strategy that shows potential in addressing the obstacles present in the administration of OVT [78].

## 4. Future Directions

As evidenced by preclinical studies, OVT is a promising development that may improve the way OS is managed. As a budding therapy, there are several opportunities for investigating treatment optimization. Establishing a safe and effective delivery method is imperative. As noted above, effective OVT delivery must address the challenges of difficult access for deep intratumoral injection, premature neutralization by the immune system, and the safety profile of increased viral loads necessary to combat pre-existing antibodies.

Combination therapies involving OVs may also warrant further exploration. The existing data on combination therapies in OS focus on only a subset of the available therapeutics. Further investigations on combination with other forms of chemotherapy and radiation therapy are needed, especially since this is standard care for unresectable OS.

Lastly, the progression of OVT to clinical trials for OS will provide valuable data that may be more widely applicable to human subjects. While murine models have shown promising results, clinical trials will highlight patient response to OVT and validate its safety and efficacy. These trials will hopefully establish OVT as a viable treatment option for OS, improving patient outcomes.

## 5. Conclusions

OVT represents a novel approach to the treatment with OS with the ability to selectively target and destroy cancer cells while sparing normal tissues. The synergistic potential of combining OVT with existing therapies represents a novel avenue for more effective treatment of OS. By addressing current challenges, this therapy could revolutionize the management of OS and overcome the seemingly inertial stagnation of survival rates for OS exhibited by the last few decades.

## Figures and Tables

**Figure 1 viruses-16-01139-f001:**
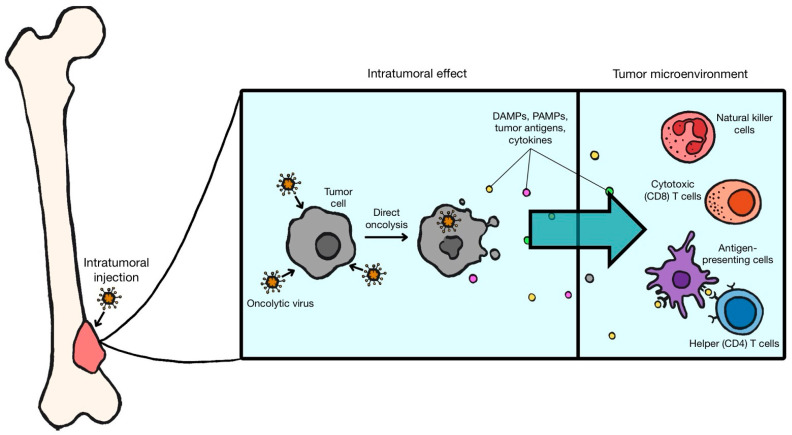
Proposed mechanism of action of OVT: oncolytic viruses selectively infect and lyse tumor cells, releasing damage-associated molecular patterns (DAMPs), pathogen-associated molecular patterns (PAMPs), tumor antigens, and cytokines. These molecules are recognized by members of the innate immune system, such as natural killer cells and antigen-presenting cells, and by members of the adaptive immune system, such as helper (CD4) and cytotoxic (CD8) T cells. This results in a robust immune response, leading to the infiltration of the tumor microenvironment by additional immune cells.

**Table 1 viruses-16-01139-t001:** Overview of oncolytic viral therapies in osteosarcoma.

Virus (+Additional Therapy)	Model	Key Findings
**Adenovirus**		
Delta-24-ACT [16]	Human cells	Potent anti-OS effect; triggered release of damage-associated molecular patterns
	Mice	Effective against primary cancer and metastases; increased survival time; safe, no toxicity
VCN-01 [17]	Human cells	Potent anti-OS effect
	Mice	Effective against primary cancer and metastases; safe, no toxicity
dlE102 + G-CSF [18]	Mice	Reduced tumor size; Increased survival with G-CSF
OBP-702 + Doxorubicin [19]	Human cells	Potent anti-OS effect with doxorubicin in a dose-dependent manner
	Mice	Combination treatment mitigated tumor growth; monotherapies were ineffective
**Herpes Simplex Virus**		
HSV1716 (Seprehvir) + Anti-PD-1 [20]	Mice	Ineffective in less immunogenic OS; Combination therapy delayed growth and prolonged survival in highly immunogenic OS
**Measles Virus**		
MV [21]	Human cells	Potent anti-OS effect
	Mice	Effective against primary cancer and metastases; increased survival
**Protoparvovirus**		
H-1PV [22]	Human cells	Potent anti-OS effect; no toxicity to normal cells
**Reovirus**		
RV (Reolysin) + Cisplatin ^1^ [23,24,25]	Human	Safe with little toxicity; no tumor response
	Human cells	Potent anti-OS effect
	Mice ^1^	Potent anti-OS effect; prevented growth; reduced size with cisplatin
	Canine cells	No anti-OS effect
**Vesicular Stomatitis Virus**		
VSV-miRNA143 [26]	Human cells	Potent anti-OS effect, greater than unmodified VSV

^1^ For RV, combination therapy was only analyzed in mice.

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
