# Peer review of "Oncolytic Viral Therapy in Osteosarcoma"

_viruses, 2024, doi:10.3390/v16071139_

Round 1

Reviewer 1 Report

Comments and Suggestions for Authors

Review of Manuscript “Oncolytic Viral Therapy in Osteosarcoma“ by Thomas Karamidas et al.. 

In their short review the authors present a compilation of the oncolytic viruses (OV) bearing potential for the treatment of osteosarcoma (OS) including adenoviruses, herpes simplex virus, measles virus, parvovirus and vesicular stomatitis virus. Classical treatment methods for OS such as surgical resection and chemotherapy have rather limited efficiency for the treatment of metastatic OS, making the application of OV’s an option in these cases. Both the therapeutic potentials of the individual OV’s in vitro and in vivo in animal models as well as major challenges and hurdles presently limiting translation into the clinical setting are described. In general, the review is well-written and provides a nice overview on this promising therapy option for metastatic OS. In some sections the molecular mechanisms should be explained in more detail, as partially outlined below.   

Major points:

1) The individual elements shown in figure 1 (e.g. different cell types) should be labeled and described in more detail for a better understanding.

2) The section 3.4 (Host vs virus) is held very general and more examples with detailed description should be provided. 

Minor points:

1) Description adenoviral OV VCN-01 (table 1 and lines 94 following): Altered cell tropism after modification of fiber shaft should be described.    

2) In table 1, Protoparovirus should read Protoparvovirus

3) Line 85 and following: The molecular modes of action of the immune stimulatory factor 4-IBBL should be described in more detail.

4) Line106 Typo: “off“ should probably read “on“.

5) Line 116: double mentioning of micro-RNA (micro-RNA and miRNA143)

6) Line 119: Number of miRNA missing

7) Line 149: The neurovirulence gene deleted in HSV1716 should be named more precisely.

8) Line 150: Mode of action of immune checkpoint receptor PD-1 should be described in more detail

9) Line 150: Is reference 46 correct?

10) Line 199: Human antibodies against reovirus are mentioned, but reovirus has not been a subject in the other parts of the manuscript. 

Reviewer 2 Report

Comments and Suggestions for Authors

The article " Oncolytic Viral Therapy in Osteosarcoma" submitted for review is focused on understanding the mechanisms by which viruses kill osteosarcoma cells, the barriers to successful viral delivery and penetration into tumor cells, the role of the immune system in viral oncolysis, strategies of the therapies and generating stronger target specific and replication viruses. Dr. Karadimas et al. summarize the supportive evidence and the current findings regarding oncolytic virotherapy within the context of osteosarcoma along with the specific challenges it may face.

The abstract of the article reflects the issues the article addresses. The summary of the article is competently designed and fully reflects the content of the article.

The article reflects the relevance of the research, contains references to journal articles on the current state of the problem.

The text of the article comprehensively reflects the stated purpose to consider the current possibilities and prospects of oncolytic virotherapy in contest of osteosarcoma. However, current challenges encountered by researchers in preclinical and clinical studies are described superficially, mainly cite other reviews and, thus, lack of corresponding references on the current research studies. It should be rewritten.

The future directions and conclusion of the article clearly correspond to the stated goals and objectives of the study.

However, I have several questions and comments regarding this manuscript. 

Major points: the figure 1 is not informative at all. The capture should describe all the elements shown in the picture. Furthermore, the figure 1 is not cited in the text. Thus, the figure 1 should be completed or deleted.

Table 1 is a mess. It should be structured and the informative capture should be reformulated.

Thus, the article is recommended for publication after the major revision.

Reviewer 3 Report

Comments and Suggestions for Authors

The review by Karadimas et al. focuses on oncolytic viral therapy as an approach to osteosarcoma treatment. There are several cancer entities, OS being among them, in which the oncolytic virotherapy approach seems presently to remain underinvestigated. Therefore, summarizing the available – although scarce - knowledge may be very helpful to achieve better understanding of both the current state-of-the-art and future directions. From that point of view, such review may trigger significant interest and potentially represents an intriguing contribution to both oncolytic virotherapy research and osteosarcoma clinical management fields.

However, in my opinion, the structure of the present review needs to be significantly improved. One major concern is that general information on OVs and OVT (basic principles of selectivity, oncolysis, immune response induction, etc.) is mixed with data on OVT applications in OS, but also in other cancer types (e.g. pancreatic cancer on line 109). My suggestion to the authors would be to (1) make a general (short) introduction to OVs and OVT as novel immunotherapeutic approach, then (2) to outline the major achievements of OVT in cancer modalities other than OS, and finally (3) focus on OV applications in OS. Paragraph 104-112 can be given as an example. In this paragraph, the authors speak about two parvovirus NS-truncated variants, which showed increased infectivity for pancreatic cancer cells. First, what is the reason of mentioning in particular pancreatic cancer here? Each OV has indeed been basically tested in multiple cancer types, but are the authors sure that the above variants (and not only the wild-type virus as indicated further - on line 109) have been studied also in OS? If yes, please add the corresponding reference. Please not that it is claimed that “Variants of H-1PV have been created to improve efficacy of OVT for OS, based off (?)  results of H-1PV…” (lines 106-107).

Oncolytic reoviruses have also been tested in the context of OS. There are clinical trials investigating reovirus safety and efficacy in OS/Ewing sarcoma patients. These studies should be undoubtedly also mentioned. Moreover, the authors say that the majority of humans have nAbs against reoviruses (line 199) but reoviruses are neither listed in Table 1 nor discussed somewhere else in the paper.

Some inappropriate word usages should be avoided. The authors should replace “isolated therapies” (line 83) by “monotherapies”, “in isolation” (line 124) by “when applied alone” or “when applied as monotherapy”, “relatively (?) immunocompetent” (line 170) by “immunocompetent”. On line 113 it is written “one final virus …”. As mentioned above, this is misleading, since other viruses, such as reoviruses, although tested, are not included in the text.

Additional remarks:

1.       “Genetically modified” (Line 11). OVT relies on the usage of both naturally occurring and modified OVs. Please correct the definition of the approach.

2.       Line 47. Similar to point 1: not only the genetically modified OVs reshape the TME. Infection of tumor cells with a naturally occurring OV may also lead to warming-up in the TME.

3.       Line 53. “…robust immune response of OVs…” Please correct. “…immune response induced by/triggered by/elicited by…”

4.       A reference to Figure 1 is missing in the text

5.       In Table 1: please correct the typing mistake: ProtoparVovirus, not Protoparovirus.

6.       Line 199. “The majority of humans have antibodies against HSV-1 and reovirus”. What about measles virus?

7.       Last but not least, the systemic usage of “V-TEC” (line 51, line 79, line 191) raises some strong concerns since it suggests that it is not merely a typing mistake… The correct abbreviation for talimogene laherparepvec is T-VEC!

Round 2

Reviewer 2 Report

Comments and Suggestions for Authors

I am completely satisfied with the authors' edits. All comments have been corrected. Thank you for your promptness. The article can be published in the presented version without changes and edits.

Author Response

Thank you so much!

Reviewer 3 Report

Comments and Suggestions for Authors

The authors provided a revised version of their manuscript, in which they made efforts to overcome some major weaknesses present in the first version submitted. Despite that, some issues are still remaining.

1.       Two major comments to Figure 1 legend. First, please correct: PAMPS are PATHOGEN-associated molecular patterns, not pattern-associated molecular patterns! (line 80). Second, the last sentence in the legend (lines 83-85) is incorrect. The immune response against OV-infected tumor cells is a consequence of their immunogenic death. It is not that the immune response is leading to the ICD of OS cells, as it is claimed in the text.

2.       Table 1, last row (VSV): what do you mean by “VSV alone”? it is not a combinatory approach, it is a single virus modified to encode a tumor-suppressive miRNA

3.       Line 98: to promote

4.       Comment to lines 133-140. Please replace “minimal significance” on line 134 by e.g. “with preferential replication in cancer cells with activated Ras signaling”. Note that virus internalization occurs in both normal and tumor cells, it is not that “the virus is readily internalized” (line 136) only by cancer cells.

5.       Lines 186-187. For the highly professional Viruses readership, I would suggest the replacement of “preventing the immune system from attacking the body’s own cells” by “in order to prevent tissue damage and maintain self-tolerance”.

6.       Line 203. Varicella is not a virus, it is a disease! The style used by the authors may be acceptable in spoken language; however, not in a specialized research article. Please use the correct virus name, and that is varicella-zoster virus. Further on line 205: “…viruses such as herpesvirus…” Again, herpesvirus is not precise. Please specify here that you mean the herpes simplex virus type 1. To note, VZV is also a herpesvirus…

7.       Line 222. Could you please explain what you mean by “wild viruses”?

8.       Line 224. Same as above: what is meant under “normal” circumstances? Which circumstances of OV administration are considered abnormal?

9.       Lines 224-226. This sentence is indeed citing the work of Li et al. However, I strongly disagree with its message. It is not true that all OVs cause such severe side effects in human patients! There are OVs that are excellently tolerated. So please correct: “…administration of some OVs…)
